# OpenReview forum: "Playing Large Games with Oracles and AI Debate"
_ICML.cc/2024/Workshop/Agentic_Markets — Agentic Markets @ ICML'24 Oral_

### Official Review · Reviewer_RLWr · 2024-06-12
**Oracle-Based Algorithms for Managing Large Decision Spaces in Language-Based Games**

**Rating:** 7
**Confidence:** 1

**Review:**

This paper explores how to manage large decision spaces in language-based games using oracles. Specifically, the authors propose oracle-based algorithms designed to efficiently minimize both internal and external regret, even in games with a very large number of possible actions. They use AI debates as a proof of concept to demonstrate the effectiveness of these algorithms.


### Pros:
- The paper presents novel algorithms to handle very large decision spaces in games using oracles.
- The developed algorithms are backed by rigorous proofs, and thorough experiments have been conducted to validate the findings.
- The work is theoretically sound and practically relevant.
- The abstract clearly outlines the purpose, methods, and contributions of the paper.
- The paper fits well within the scope of ICML and the workshop. It addresses central topics in AI and game theory, making it highly relevant to the field.
- The practical implications, as well as the highlighted future works and limitations in the paper, provide room for discussion and interactive engagement during the workshop.

### Cons:
- The paper is highly technical and may be difficult to understand for non-specialists or those new to the field.

---

### Official Review · Reviewer_GBBP · 2024-06-12
**Simultaneous internal and external regret minimization**

**Rating:** 7
**Confidence:** 2

**Review:**

The paper studies regret minimization in two-players general-sum repeated games with large action space. Given the large action space, the authors focus on oracle-based solutions. They consider two, known in the literature, types of regret; external regret and internal regret. Their main contribution is Algorithm 1 which achieves $InternalRegret \le O(\sqrt{T \ln N})$ and $ExternalRegret \le O(\sqrt{T \ln N})$, by making in each step calls to a smooth optimization oracle. The algorithm's running time is $poly(T)$

**Comments - Questions**
- For someone who is not expert in the area, it would be helpfulto elaborate more on the differences between the pure optimization oracle and the smooth. Does the latter introduce any additional computational overhead compared the former one?
- What would be the difference in the regret bounds between your "simultaneous internal and external regret minimization" algorithm and the reduction from external to internal regret by Blum and Mansour (2017)
- The experimental setting does not explicitly align with the theoretical framework, since you consider only zero sum games. Could you expand on the challenges of inestigating empirically general sum games?
- Does the different bounds on internal regret in Table 1 depend on you using smooth optimization oracles. Could you expand on that?